# The Biologically Active Compounds in Fruits of Cultivated Varieties and Wild Species of Apples

**DOI:** 10.3390/molecules30193978

**Published:** 2025-10-04

**Authors:** Alexander A. Shishparenok, Anastasiya N. Shishparenok, Heather A. Harr, Valentina A. Gulidova, Eugene A. Rogozhin, Alexander M. Markin

**Affiliations:** 1Petrovsky National Research Centre of Surgery, 119435 Moscow, Russia; 2Institute of Biomedical Chemistry, 119121 Moscow, Russia; a.shishparyonok@yandex.ru; 3Independent Researcher, 664033 Irkutsk, Russia; 4Department of Agricultural Technologies, Storage and Processing of Agricultural Products, Ivan Bunin Yelets State University, 399770 Yelets, Russia; 5Shemyakin–Ovchinnikov Institute of Bioorganic Chemistry, Russian Academy of Sciences, 117997 Moscow, Russia; rea21@list.ru; 6All–Russian Institute of Plant Protection, 196608 St. Petersburg–Pushkin, Russia; 7Department of Histology, Petrovsky Medical University, 119435 Moscow, Russia; 8Department of Histology, Cytology and Embryology, Medical Institute, Peoples’ Friendship University of Russia Named After Patrice Lumumba (RUDN University), 117198 Moscow, Russia

**Keywords:** apples, phenolic compounds, organic acids, pigments, triterpenoids, sweetness, health

## Abstract

Insufficient fruit intake is a major contributor to the development of non-communicable diseases, as the global average of daily fruit consumption remains far below the recommended levels. Apples are among the most widely consumed fruits worldwide, making them an ideal target for nutritional enhancement. Enhancing the content of health-promoting compounds within apples offers a practical way to increase bioactive intake without requiring major dietary changes. This review evaluates which of the 41 biologically active compounds considered in this article can reach physiologically relevant intake levels at the current average daily consumption of cultivated and wild apples. Comparative analysis shows that wild apples consistently contain higher concentrations of phenolic compounds and organic acids than cultivated varieties, in some cases by more than tenfold. At the average daily fruit intake of 121.8 g, wild species provide effective doses of epicatechins, anthocyanins, chlorogenic acid, and malic acid. In contrast, cultivated apples reach this level only for chlorogenic acid. Notably, less than 50 g of wild apple is sufficient to supply physiologically relevant amounts of several polyphenols. These findings highlight the potential of wild apple species as donors of bioactive compounds and provide a framework for breeding future apple cultivars that combine consumer appeal with enhanced health benefits.

## 1. Introduction

Daily consumption of less than 800 g of fruits and vegetables has been associated with increased mortality from non-communicable diseases. The recommended daily intake is 400 g of fruits and 400 g of vegetables, based on the content of bioactive compounds with proven health benefits, which each has. Nevertheless, the actual global average fruit consumption remains only 121.8 g per person per day, substantially below the recommended intake [1]. This low consumption is explained mainly by high fruit prices and entrenched dietary habits [2]. A feasible strategy to improve bioactive compound intake is not merely to increase fruit consumption but to enhance commonly consumed fruits with these compounds, ensuring that effective doses can be achieved even at current consumption levels [3]. Among fruits, apples are highly accessible and widely consumed, making them a suitable target for enrichment [4].

Apples are a rich source of bioactive compounds, including polyphenols, flavonoids, phenylpropanoid derivatives, organic acids, pigments, triterpenoids, and fatty acids [5,6,7,8] (Figure 1). However, the amount of bioactive compounds varies substantially among apple genotypes. Wild apple species typically produce small, sour-bitter fruits because of elevated levels of organic acids and tannins, while also exhibiting high concentrations of health-promoting bioactive compounds. In contrast, cultivated varieties contain lower levels of these compounds but are larger and sweeter, reflecting selective enhancement of the sugar-to-acid balance during domestication [9]. Determining which bioactive compounds are present in wild and cultivated apple species, comparing their concentrations, and identifying those sufficient to meet health-promoting levels at current global consumption are critical steps toward understanding how to improve cultivated apple varieties.

Despite the potential of wild apple species to enhance the health-promoting properties of cultivated varieties, it remains unclear which bioactive compounds can realistically reach effective doses at the average daily apple consumption of 121.8 g. Existing studies are fragmented, with some focusing on the effects on human and animal health, others on the biochemical composition of fruit, effective doses of metabolites, or the impact of whole-fruit consumption. To date, no study has systematically integrated these datasets to determine which compounds from wild and cultivated apples can achieve physiologically adequate levels at the current global average daily consumption of apples.

Based on available data, most bioactive compounds are present at higher levels in wild apple species. Therefore, we hypothesize that wild apples may provide sufficient quantities of certain bioactive compounds to achieve effective doses at the current average consumption level, and that these compounds could be utilized in breeding programs to enrich cultivated varieties. The objective of this review is to assess which bioactive compounds can be obtained at physiologically effective doses through the average daily consumption of 121.8 g of cultivated and wild apples [10,11,12,13,14,15,16].

Publications were collected from PubMed, Scopus, and Web of Science using keywords including “polyphenols,” “organic acids,” “pigments,” “fatty acids,” “triterpenoids,” and “effective doses.” Inclusion criteria included the following: (1) studies reporting concentrations of these compounds in cultivated or wild apples; (2) studies confirming their beneficial properties; (3) studies providing effective dose data; and (4) the use of established analytical techniques such as HPLC, GC/MS, or spectrophotometry. This data was used to construct comparative tables of cultivated and wild apple species. Next, bioactive compound concentrations were divided by their reported effective doses, and results were expressed as the equivalent fruit mass required to satisfy human physiological needs.

The findings will identify which bioactive compounds are present in sufficient quantities in wild apple species at the current global average daily consumption. This information can support breeding strategies aimed at enhancing cultivated apple varieties for specific bioactive compounds using wild germplasm, ultimately improving the nutritional quality of apples without compromising taste or size.

## 2. Effects of Apple and Apple Extract Consumption on Human Diseases

A plethora of biologically active compounds have been identified in apples and apple extracts, which exert a wide range of effects on human health [17]. A 15-year longitudinal study was conducted to investigate the relationship between apple consumption and all-cause mortality in 1456 women over the age of 70. The results indicated a 35% decrease in mortality among individuals who consumed an apple daily [17]. The observed decline in overall mortality among apple consumers can be attributed to a decrease in the probability of mortality from specific diseases. Cardiovascular diseases represent the most prevalent type of disease in the global population [18]. Studies have demonstrated that regular consumption of apples plays a significant role in the primary prevention of cardiovascular diseases, including atherosclerosis, coronary heart disease, and myocardial infarction [19]. Furthermore, consuming apples and apple juice has been shown to play a role in preventing various types of cancer, including lung, colorectal, oral cavity, and breast cancers. For instance, a study conducted on a female population demonstrated that consuming 57 g of apples on a daily basis was associated with a reduced risk of developing breast cancer. A single consumption of one liter of apple juice was found to exert an antigenotoxic effect on the intestine [20,21]. Despite the high sugar content in apples, ingesting them has been associated with a reduced risk of developing diabetes. The positive impact on diabetes prevention is attributed to the presence of soluble fibers and biologically active compounds found in apple fruit [17,19].

A recent study found that consuming apple products over six weeks led to a decrease in plasma concentrations of C-reactive protein (CRP), interleukin-6 (IL-6), and LPS-binding protein (LBP). Therefore, regular apple consumption can reduce the prevalence of asthma, particularly in children [17]. The impact of apple consumption on the human gut and microbiome is a particularly salient subject. Apples contain probiotics, which utilize beneficial bacteria, particularly bifidobacteria, in the gastrointestinal tract to produce short-chain fatty acids (SCFAs) [22,23,24,25]. The ingestion of a minimum of 100 **g** per day of whole apples has been shown to enhance cerebral blood flow through the production of nitric oxide, thereby promoting improvements in mood and cognitive function. Additionally, the production of nitric oxide has been shown to enhance endothelial function, which, in turn, has been associated with reductions in blood pressure and the prevention of chronic kidney disease [21,26,27]. The presence of potassium and calcium within apple fruits suggests that their consumption may improve bone health [28]. The beneficial effects of apples on human health are caused by individual bioactive compounds; thus, it is pertinent to analyze both the compound classes and the specific metabolites in the subsequent section.

## 3. Description of the Beneficial Properties of Biologically Active Compounds

### 3.1. Polyphenolic Compounds

Phenolic compounds and their various derivatives, including glycosides, are defined as compounds that contain aromatic rings with a hydroxyl group. Should the aromatic ring of phenolic compounds contain more than one hydroxyl group, the resulting compounds are referred to as polyphenols [29].

The polyphenolic compounds present in apple fruit include chlorogenic acid, caffeic acid, cinnamic acid, catechins, procyanidins, phloretin, phlorizin, quercetin, anthocyanins, gallic acid, vanillic acid, ferulic acid, p-coumaric acid, rutin, kaempferol, 3-hydroxyphloridzin, p-coumaroylquinic acid, protocatechuic acid, neochlorogenic acid, hyperoside, and myricetin. The total phenolic content is less important for humans than the content of certain individual polyphenolic compounds. Some of these compounds have a beneficial effect on human health [30]. Apples are a significant source of secondary plant metabolites, which are vital for human nutrition. In this study, 21 polyphenolic compounds were selected for detailed analysis. Collectively, these compounds exhibit 25 distinct health-promoting properties, including antioxidant/anti-inflammatory, antitumor/anticancer, antidiabetic, hypoglycemic, cardioprotective, hypocholesterolemic, hypotensive, anti-atherosclerosis, hepatoprotective, nephroprotective, glomerular regeneration, immunostimulating, anti-allergenic/immunomodulatory, neuroprotective, anti-rheumatic, bone-protective/osteoprotective, analgesic, dermatoprotective, wound healing, hair growth-stimulating, anti-obesity/metabolic regulation, antibacterial, antiviral, anti-aging, and membrane-modulating/drug-permeability-enhancing [31,32,33,34,35,36,37,38,39,40,41,42,43,44,45,46,47,48,49]. Among the analyzed phenolic compounds, phlorizin demonstrated the broadest spectrum of health-promoting effects, being associated with 15 distinct properties. Chlorogenic acid exhibited 11 beneficial activities, while rutin was linked to 10 health-related effects. A comprehensive overview of the health-promoting properties associated with each individual polyphenol is provided in Appendix A. The main phenolic compounds in apple fruits are shown in Figure 2.

### 3.2. Organic Acids

Organic acids are organic compounds with weak acidic properties. The classification of organic acids is based on the number of hydroxyl or carboxyl functional groups [50]. The three main acids found in apples are malic, citric, and ascorbic [51] (Figure 3). In the present study, four organic acids were selected for detailed examination. Together, these metabolites were associated with 21 distinct health-promoting effects, including antioxidant/anti-inflammatory, antitumor/anticancer, antidiabetic, hypoglycaemic, cardioprotective, hypocholesterolemic, hypotensive, anti-atherosclerosis, nephroprotective, anti-allergenic/immunomodulatory, neuroprotective, anti-rheumatic, bone-protective/osteoprotective, dermatoprotective, anti-obesity/metabolic regulation, antibacterial, antiviral, anti-aging, saliva stimulation/xerostomia therapy, embryoprotective, and anti-scorbutic [52,53,54,55,56,57,58,59,60,61,62,63]. Among the organic acids selected for analysis, ascorbic acid exhibited the broadest spectrum of biological activities, demonstrating 12 distinct health-promoting properties. For a more detailed overview of the health-promoting properties attributed to each organic acid, readers are referred to Appendix A.

### 3.3. Pigments

Plant pigments are a class of secondary metabolites that perform a variety of vital functions in plants. Pigments are involved in photosynthesis, metabolism, growth, and protection against photooxidative damage. Furthermore, pigments play a pivotal role in determining the coloration and attractiveness of fruits [64,65]. The primary pigments present in apple fruits are chlorophylls and carotenoids (Figure 4). Together, these metabolites demonstrate 12 distinct health-promoting activities, including antioxidant/anti-inflammatory, antitumor/anticancer, antidiabetic, hypoglycemic, cardioprotective, anti-atherosclerotic, hepatoprotective, anti-allergenic/immunomodulatory, neuroprotective, dermatoprotective, anti-aging, and ocular-protective [66,67,68,69,70,71]. Carotenoids exhibit a broader range of health-promoting properties compared to chlorophylls. Specific information on the health-promoting properties of chlorophylls and carotenoids is provided in Appendix A.

### 3.4. Triterpenoids

Triterpenoids constitute a group of secondary plant metabolites found in cuticular waxes. A recent study has proven that triterpenoids produce pharmacological effects on human health [72] (Figure 5). The following triterpenoids have been identified in apple fruits: ursolic acid, 3-oxo-23-hydroxyurs-12-en-28-oic acid, corosolic acid, annurcoic acid, betulinic acid, euscaphic acid, maslinic acid, pomolic acid, and pomaceic acid. Collectively, these nine triterpenoids exhibit 23 distinct health-promoting properties, including numerous effects against infectious agents and parasites. Among the various activities of these triterpenoids, the following were highlighted by the sources referenced while researching: antioxidant/anti-inflammatory, antitumor/anticancer, antidiabetic, hypoglycaemic, cardioprotective, hypocholesterolemic, hypotensive, anti-atherosclerosis, anti-allergenic/immunomodulatory, neuroprotective, anti-rheumatic, bone-protective/osteoprotective, analgesic, dermatoprotective, anti-obesity/metabolic regulation, antibacterial, antiviral, anti-aging, myoprotective/muscle-enhancing, anthelmintic, antimalarial, insecticidal, and anti-fatigue [73,74,75,76,77,78,79,80,81,82,83,84]. Ursolic acid and its derivatives exhibit the greatest number of health-promoting properties, with a total of 11 distinct activities. In addition to general health-promoting properties, betulinic acid also exhibits antibacterial, antiviral, anthelmintic, and antimalarial activities [80]. Information on the health-promoting properties of each of the nine triterpenoids can be found in Appendix A.

### 3.5. Fatty Acids

Fatty acids are components of lipid compounds and are composed of an acyl (hydrocarbon) chain accompanied by methyl and carboxyl groups at opposing ends. Fatty acids can be classified into three distinct groups based on their structural characteristics and molecular composition: saturated fatty acids, which lack any double bonds; monounsaturated fatty acids, which possess a single double bond; and polyunsaturated fatty acids, which contain two or more double bonds. In the case of apple fruits, fatty acids can be found in the peel of the fruit and in the seeds. The predominant fatty acids present in apples are illustrated in Figure 6 [85]. Linoleic and oleic acids are found in high concentrations within the seeds of apple fruits [85]. Linoleic and oleic acids exhibit 11 distinct health-promoting properties for humans, including antitumor/anticancer, antidiabetic, cardioprotective, hypocholesterolemic, hypotensive, neuroprotective, anti-rheumatic, bone-protective/osteoprotective, dermatoprotective, hair-protective, and reproductive health protective [86,87,88,89]. Appendix A provides a detailed overview of the beneficial properties associated with linoleic and oleic acids.

## 4. A Comparative Analysis of the Biochemical Composition of Cultivated and Wild Apple Species

After conducting a literature analysis, 41 biologically active compounds and their groups were selected for comparative analysis (see Table 1, Figure 7 and Figure 8). Malic acid, citric acid, anthocyanins, chlorogenic acid, epicatechins, gallic acid, procyanidins, quercetins, catechins, and ascorbic acid are basic compounds in apple fruits.

The biologically active compounds were categorized to elucidate the substantial disparities in the levels of these compounds between cultivated varieties and wild apple species. As illustrated in Figure 7, the differences in content between cultivated varieties and wild apple species exceeded 200%. Conversely, Figure 8 illustrates differences falling below 200%. Twenty-eight biologically active compounds were predominant in the fruits of wild apple species. The most pronounced disparities were observed in the amount of anthocyanins. Specifically, the levels of anthocyanins in wild apple fruits were 103.88 times higher than those in cultivated varieties. In addition, the following compounds are predominant in the fruits of wild apple species: gallic acid (30.82 times), citric acid (26.77 times), epicatechins (23.73 times), hyperoside (12.62 times), total phenolic content (10.29 times), ferulic acid (7.82 times), vanillic acid (5.82 times), 3-hydroxyphloridzin (5.82 times), caffeic acid (5.79 times), p-coumaric acid (3.71 times), neochlorogenic acid (3.28 times), rutin (3.21 times), p-coumaroylquinic acid (3.13 times), total organic acids (2.93 times), carotenoids (2.37 times), phlorizin (2.37 times), catechins (2.16 times), quercetins (2.12 times), chlorophylls (2.12 times), ascorbic acid (2.02 times), chlorogenic acid (1.92 times), procyanidins (1.87 times), myricetin (1.78 times), malic acid (1.66 times), linoleic acid (1.56 times), euscaphic acid (1.23 times), and oleic acid (1.01 times). Conversely, in fruits of cultivated varieties, there is a higher prevalence of phloretin (31.7 times), cinnamic acid (12.75 times), pomaceic acid (2.55 times), annurcoic acid (1.78 times), 3-oxo-hydroxy-urs-12-en-28-oic acid (1.71 times), pomolic acid (1.40 times), corosolic acid (1.28 times), betulinic acid (1.07 times), quinic acids (1.03 times), maslinic acid (1.03 times), and ursolic acid (1.03 times).

The most significant deviations in levels of biologically active compounds are observed in wild apple species. A significant proportion of the biologically active compounds exhibited a deviation of more than 200% from the cultivated apple variety, with 14 out of 16 compounds demonstrating such deviations.

The fruits of cultivated varieties contain several biologically active compounds in greater quantities than wild apple species. However, the concentration of these biologically active compounds in cultivated varieties differs from that of wild apple species by less than 200%. Consequently, the utilization of wild apple species to modify cultivated varieties is expected to yield a more pronounced enhancement in the biochemical composition of the fruit (Figure 8).

The authors conducted a review of the available literature to ascertain the effective doses of biologically active compounds and their levels in the fruits of cultivated and wild apple species (Table 1). The analysis revealed considerable variation in the levels of biologically active compounds across different apple varieties. This variation highlights the importance of determining which specific compounds make apples a meaningful dietary source.

To identify these key compounds, it is necessary to express their content in terms of the fruit mass required to provide an effective dose (Table 2). This approach involves dividing the content of biologically active compounds in the fruits of cultivated and wild apple species by the corresponding effective doses. Detailed calculations, including formulas and unit conversions, are provided in the Appendix A, ensuring a transparent and reproducible methodology.

Once the fruit masses required to achieve an effective dose have been determined, the next step is to assess the feasible daily intake of apples without disrupting a balanced diet. Epidemiological studies indicate that daily consumption of fruits and vegetables, whether combined or separately, up to 800 g, reduces mortality from most diseases, with the exception of cancer, for which the optimal intake is up to 600 g. Intake above 800 g does not confer additional benefits. In practice, however, the global average fruit consumption in 2021 was only 121.8 g per person per day [1]. Therefore, when evaluating cultivated and wild apple species as sources of specific biologically active compounds, it is reasonable to consider 121.8 g/person/day as the practical upper limit of daily fruit intake.

Analyzing the values presented in Table 2, which indicate the estimated fruit mass required to ensure an adequate intake of biologically active compounds, reveals notable differences between cultivated and wild apple species. A portion of 121.8 g of wild apple fruits provides effective doses of a wider range of biologically active compounds compared with the same portion of cultivated varieties. Cultivated apples primarily satisfy the requirement for chlorogenic acid, whereas wild apple species cover epicatechins, anthocyanins, chlorogenic acid, and malic acid. Notably, less than 50 g of wild apple fruits are sufficient to deliver effective doses of epicatechins, anthocyanins, and chlorogenic acid—roughly half of the actual average daily fruit consumption. These findings suggest that incorporating wild apple species into the breeding of cultivated varieties could be an effective strategy for enhancing their levels of epicatechins, anthocyanins, and malic acid [1].

Three biologically active compounds for which an effective dose can be obtained from less than 50 g of wild apple fruits are polyphenols (Table 2). To identify the most promising wild apple species for potential breeding programs, a comparative analysis was conducted (Table 3). The study presents the total polyphenol content across different wild apple species. Based on this data, the wild species *Malus prunifolia* emerges as a strong candidate for enriching cultivated varieties with polyphenols. Notably, the genome of *M. prunifolia* was sequenced in 2022 [167]. Genome analysis can help identify genetic loci responsible for the synthesis of polyphenols that provide a wide range of health benefits without imparting the tart taste associated with tannins. This approach can preserve the flavor of cultivated apples while incorporating the advantageous traits of wild apple species [168].

## 5. Conclusions

This review integrates current knowledge on the health-promoting properties of apples by systematically combining data on the content of 41 biologically active compounds and their groups with reported effective doses. Such an approach allows not only the characterization of the phytochemical diversity of apple fruits but also the identification of metabolites that can realistically contribute to human health through regular consumption.

The comparative analysis revealed pronounced differences between cultivated and wild apple species. Wild species consistently demonstrated higher concentrations of phenolic compounds, triterpenoids, organic acids, fatty acids, and pigments. Converting compound concentrations within apples into fruit mass required to reach an effective dose provided a practical tool for assessing health-promoting value. Notably, several bioactive compounds—including epicatechins, anthocyanins, chlorogenic acid, and malic acid—can reach physiologically relevant levels through realistic daily intake of wild apples. In contrast, cultivated varieties can reach such levels only for chlorogenic acid.

Among wild species, *Malus prunifolia* stands out due to its exceptionally high polyphenol content and the availability of genomic resources, making it a promising candidate for targeted use in breeding. Incorporating wild germplasm into breeding programs represents a strategic pathway to enhance the health-promoting properties of cultivated apples.

In summary, the systematic integration of biochemical data with practical dose analysis provides a foundation for nutritionally informed breeding strategies. This framework supports the development of future apple cultivars that combine consumer appeal with enhanced health benefits, thereby contributing to improved public health.

## Figures and Tables

**Figure 1 molecules-30-03978-f001:**
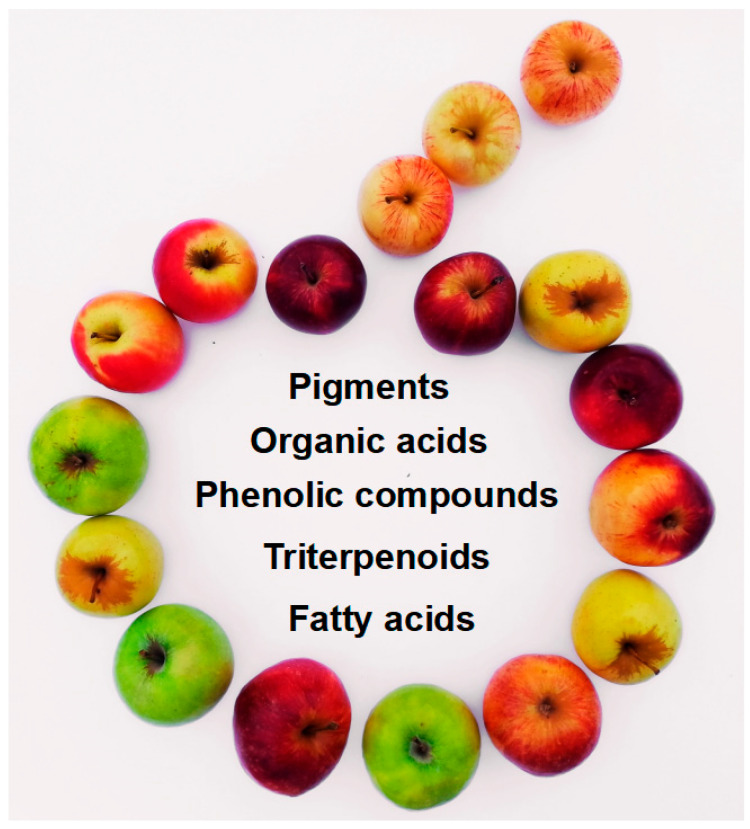
Beneficial bioactive compounds in apple fruits.

**Figure 2 molecules-30-03978-f002:**
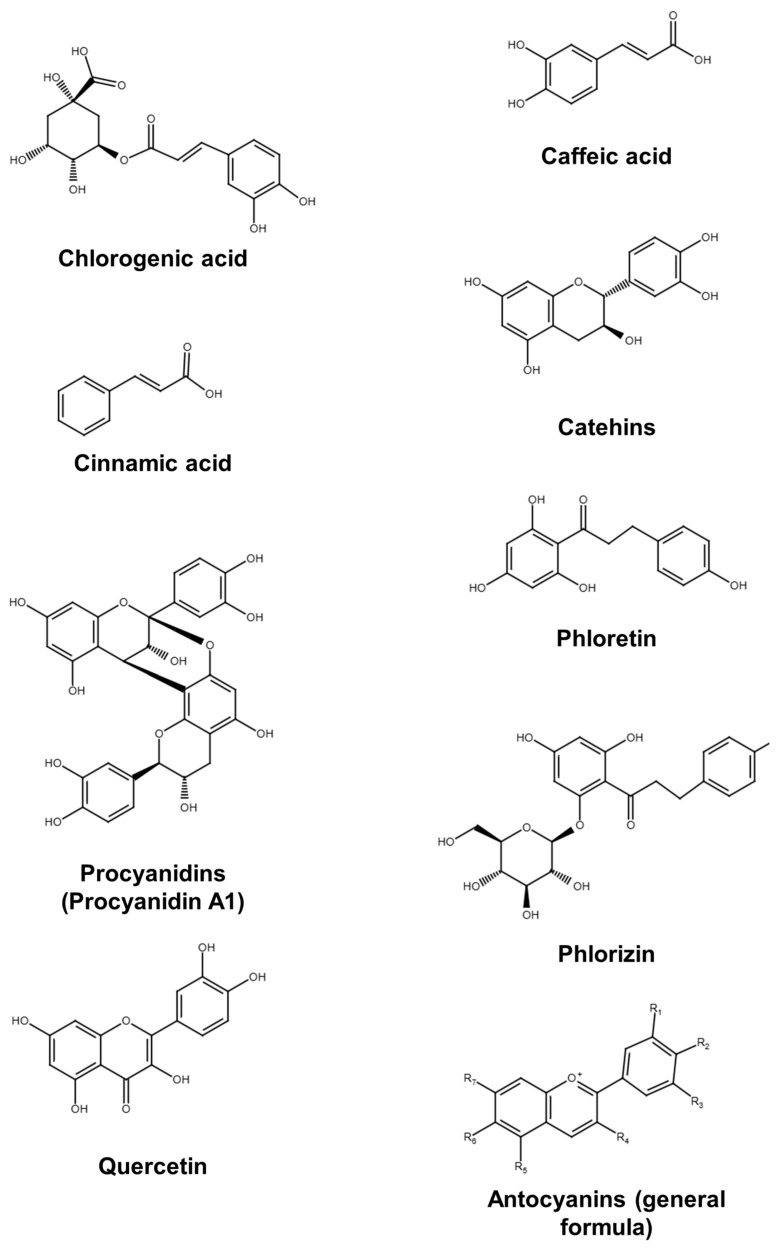
Main phenolic compounds in apple fruits.

**Figure 3 molecules-30-03978-f003:**
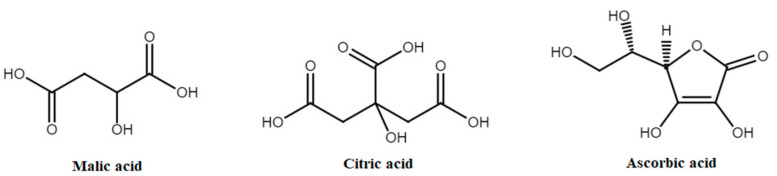
Main organic acids in apple fruits.

**Figure 4 molecules-30-03978-f004:**
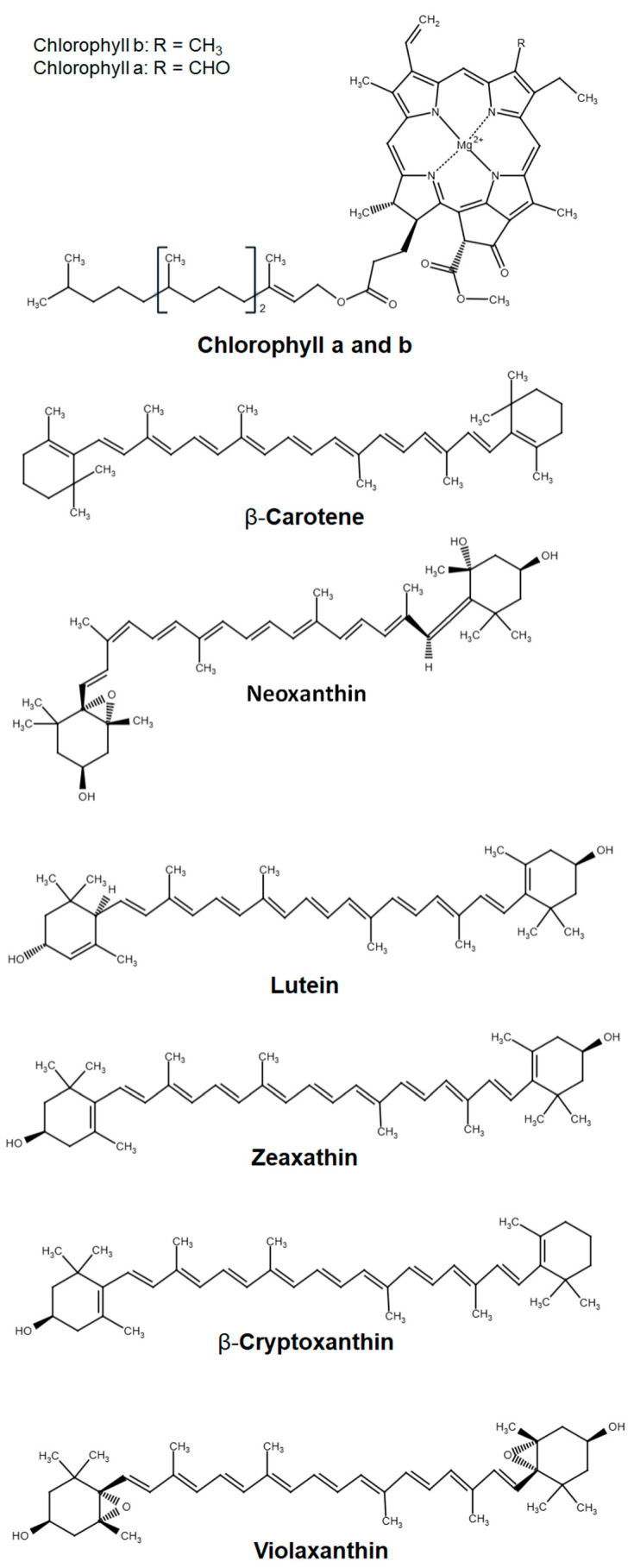
Primary pigments in apple fruits.

**Figure 5 molecules-30-03978-f005:**
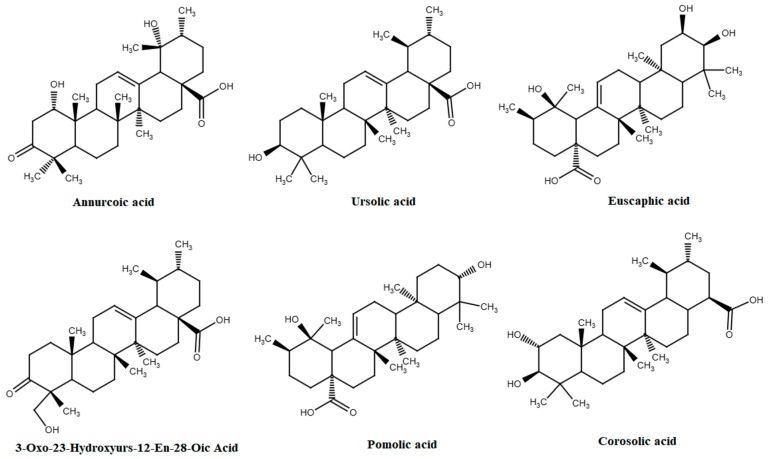
Main triterpenoids in apple fruits.

**Figure 6 molecules-30-03978-f006:**
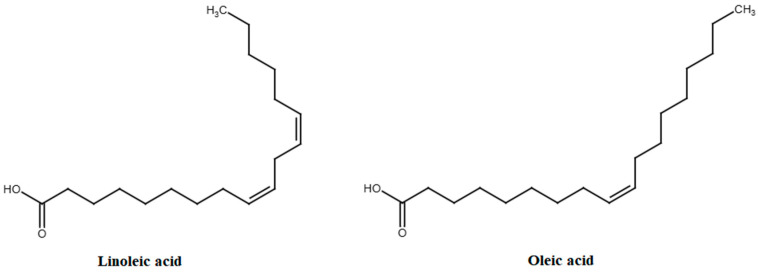
The predominant fatty acids in apple fruits.

**Figure 7 molecules-30-03978-f007:**
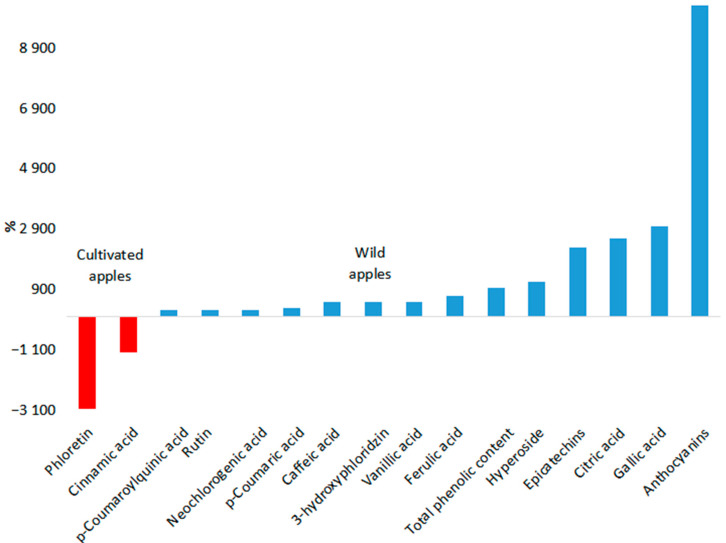
Differences in the content of biologically active compounds between wild and cultivated apple species. Differences more than 200%. Refer to Appendix A for a listing of (1) the multiple differences in the biologically active compounds between cultivated and wild apple species and (2) calculations of the percentages of these differences.

**Figure 8 molecules-30-03978-f008:**
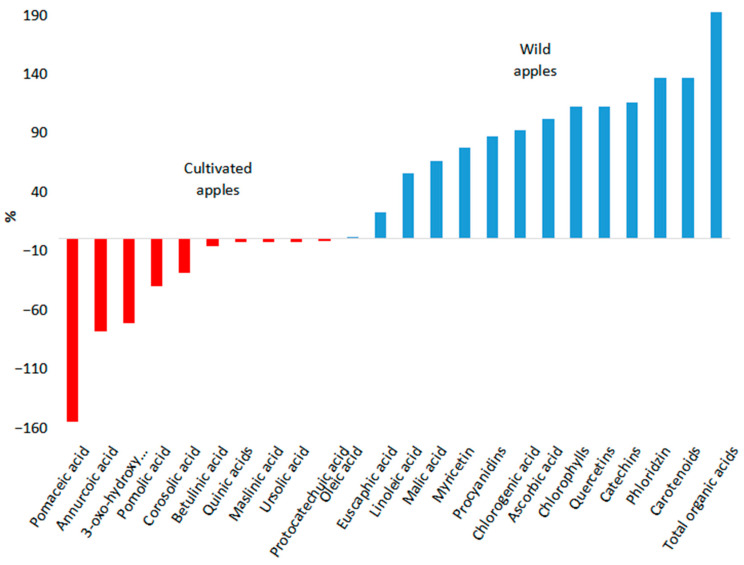
Differences in the content of biologically active compounds between wild and cultivated apple species. Differences less than 200%.

**Table 1 molecules-30-03978-t001:** General table of the effective dose and the content of biologically active compounds in wild and cultivated apples.

Chemical Compounds	Type of Apple	Chemical Compound Content, Range (Mean Value) μg/g FW	Effective Dose for Humans	Study
Phenolic Compounds				
Total phenolic content	Cultivated apples	404–2170 (1016.8) †	220 mg/day	[90,91,92,93,94]
	Wild apple species	1460.98–2.28 × 104 (1.05 × 10^4^) †		
Chlorogenic acid	Cultivated apples	28.25–1104 (141.06) †	13.5 mg/day	[95,96,97,98,99]
	Wild apple species	25.02–2.75 × 10^4^ (270.93) †		
Caffeic acid	Cultivated apples	1.1–18.88 (7.47) †‡	159.4 mg/day	[97,99,100,101,102,103,104]
	Wild apple species	0.14–138.1 (43.22) †		
Catechins	Cultivated apples	0.45–210.9 (36.06) †	592.9 mg/day	[98,102,105,106,107,108]
	Wild apple species	20.9–134.9 (77.9) †		
Epicatechins	Cultivated apples	13.13–960.4 (88.87) †	0.5 mg/kg	[96,98,102,109,110]
	Wild apple species	62.18–5816.3 (2108.98) †		
Procyanidins	Cultivated apples	57.8–895.3 (185.1) †	704 mg/day	[98,111,112,113,114]
	Wild apple species	5.9–2284.4 (345.58) †‡		
Phlorizin	Cultivated apples	5.61–115.5 (25.95) †	60 mg/day	[96,98,102,115,116]
	Wild apple species	29.48–430.4 (61.46) †		
Quercetins	Cultivated apples	23.73–408.4 (40.73) †	>500 mg/day	[96,98,117]
	Wild apple species	69.65–290.1 (86.39) †		
Anthocyanins	Cultivated apples	4–123 (35.4) †	80 mg/day	[90,111,118,119,120]
	Wild apple species	34.74–18930 (3677.37) †‡		
Gallic acid	Cultivated apples	6.3–11.4 (8.93) †	1.42 mg/kg ⱷ	[97,102,121,122]
	Wild apple species	0.91–547.2 (275.2) †		
Vanillic acid	Cultivated apples	15.03–15.55 (15.29)	6.92 mg/kg ⱷ	[115,121,123]
	Wild apple species	11.1–314.26 (88,97) †		
Ferulic acid	Cultivated apples	1.05–1.37 (1.21)	500–1000 mg/day	[115,121,124]
	Wild apple species	2.43–14.94 (9.46) †		
p-Coumaric acid	Cultivated apples	1.33–14.6 (6.41) †	1.57 mg/kg ⱷ	[97,115,121,125,126]
	Wild apple species	9.52–37.18 (23.76) †		
Rutin	Cultivated apples	38.7–44.4 (41.89) †	1000 mg/day	[97,115,127,128]
	Wild apple species	56.51–212.09 (134.3) †		
Kaempferol	Cultivated apples	1.6–86 (28.89) †	0.36 mg/kg ⱷ	[97,118,129,130]
	Wild apple species	0.8–20 (n.d.) †		
Cinnamic acid	Cultivated apples	0.02–1.25 (0.51) ‡	159.4 mg/day	[104,111]
	Wild apple species	n.d. (0.04) ‡		
Phloretin	Cultivated apples	0.21–7.52 (19.97) †‡	0.71 mg/kg ⱷ	[111,131,132,133,134]
	Wild apple species	n.d. (0.63) †‡		
3-hydroxyphloridzin	Cultivated apples	n.d. (1.1)	n.d.	[98]
	Wild apple species	n.d. (6.4)		
p-Coumaroylquinic acid	Cultivated apples	0.56–29 (9.28) †	n.d.	[105,112,135]
	Wild apple species	n.d. (29) ‡		
Protocatechuic acid	Cultivated apples	0.6–7.3 (3.43) †‡	0.55 mg/kg ⱷ	[99,121,136,137,138]
	Wild apple species	1.41–13.36 (3.37) †		
Neochlorogenic acid	Cultivated apples	0.45–10.37 (1.67) †	200 mg/day	[95,96,125,139]
	Wild apple species	n.d. (5.47) †		
Hyperoside	Cultivated apples	0.6–1.6 (0.95) ‡	3.46 mg/kg ⱷ	[99,140,141]
	Wild apple species	2.77–23.1 (11.99)		
Myricetin	Cultivated apples	5.81–12.1 (8.96)	3.56 mg/kg/day ⱷ	[115,142]
	Wild apple species	n.d. (15.93)		
Triterpenoids				
3-Oxo-23-hydroxyurs-12-en-28-oic acid	Cultivated apples	n.d. (21.9)	n.d.	[98]
	Wild apple species	n.d. (12.8)		
Annurcoic acid	Cultivated apples	n.d. (75.9)	n.d.	[98]
	Wild apple species	n.d. (42.6)		
Betulinic acid	Cultivated apples	n.d. (8.1)	1.42 mg/kg/day ⱷ	[98,143]
	Wild apple species	n.d. (7.6)		
Corosolic acid	Cultivated apples	n.d. (21.2)	10 mg/kg	[98,144]
	Wild apple species	n.d. (16.5)		
Euscaphic acid	Cultivated apples	n.d. (25.7)	3.56 mg/kg ⱷ	[98,145]
	Wild apple species	n.d. (31.6)		
Ursolic acid	Cultivated apples	n.d. (59.6)	2.65 mg/kg	[98,146]
	Wild apple species	n.d. (58.1)		
Maslinic acid	Cultivated apples	n.d. (11.9)	30 mg/day	[98,147]
	Wild apple species	n.d. (11.6)		
Pomolic acid	Cultivated apples	n.d. (21.3)	0.06 mg/kg ⱷ	[98,148]
	Wild apple species	n.d. (15.2)		
Pomaceic acid	Cultivated apples	n.d. (5.1)	n.d.	[98]
	Wild apple species	n.d. (2)		
Fatty acids				
Linoleic acid	Cultivated apples	n.d. (11.3)	20,000 mg/day	[98,149]
	Wild apple species	n.d. (17.6)		
Oleic acid	Cultivated apples	n.d. (12.4)	13,750–20,750 mg/day	[98,150]
	Wild apple species	n.d. (12.5)		
Organic acids				
Total organic acids	Cultivated apples	1629–1.01 × 10^4^ (5253) †	n.d.	[151,152]
	Wild apple species	2580–4.46 × 10^4^ (1.54 × 10^4^)		
Malic acid	Cultivated apples	1542–1.77 × 10^4^ (6966) †	1200 mg/day	[98,151,152,153]
	Wild apple species	2580–2.93 × 10^4^ (1.16 × 10^4^) †		
Ascorbic acid	Cultivated apples	10.48–220.5 (38.39) †	40 mg/day	[98,154,155]
	Wild apple species	22.07–325 (77.41) †		
Citric acid	Cultivated apples	32.9–551.2 (84.22) †	2700 mg/day	[151,152,156,157,158]
	Wild apple species	430–24210 (2254.5) †		
Quinic acids	Cultivated apples	1.5–571.9 (19.17)	5.34 mg/kg	[98,159]
	Wild apple species	1.3–287.6 (18.6)		
Pigments				
Chlorophylls	Cultivated apples	0.2–8.08 (3.07) †‡	150 mg	[160,161,162,163]
	Wild apple species	n.d. (6.51)		
Carotenoids	Cultivated apples	1.328–4.95 (15.38) †‡	6.45 mg/day	[160,161,162,163,164,165,166]
	Wild apple species	0.84–99 (36.38) †		

Note: †—Mean value based on several studies; ‡—Chemical content in apple pulp; ⱷ—Human equivalent dose; n.d.—no data. Tables containing initial data, prior to the averaging of biologically active compound content in both wild and cultivated apples, can be found in Appendix A.

**Table 2 molecules-30-03978-t002:** Estimated fruit mass (g) required to ensure an adequate intake of biologically active compounds.

Chemical Compounds	Cultivated Apple Varieties	Wild Apple Species
Epicatechins	337.6	14.2
Anthocyanins	2259.89	21.75
Chlorogenic acid	95.7	49.8
Malic acid	172.27	103.89
Carotenoids	419.38	177.30
Pomolic acid	169.01	236.84
Gallic acid	9540.87	309.59
Ascorbic acid	1041.94	516.73
Phloridzin	2312.1	976.2
Citric acid	3.21 × 10^4^	1197.60
Procyanidins	3803.3	2037.2
Maslinic acid	2521.01	2586.21
Ursolic acid	2667.79	2736.66
Caffeic acid	2.13 × 10^4^	3688.1
p-Coumaric acid	1.47 × 10^4^	3964.65
Vanillic acid	2.72 × 10^4^	4666.74
Quercetins	1.23 × 10^4^	5787.7
Euscaphic acid	8311.28	6759.49
Rutin	2.39 × 10^4^	7446.02
Catechins	2.94 × 10^4^	7611
Protocatechuic acid	9620.99	9792.28
Betulinic acid	1.05 × 10^4^	1.12 × 10^4^
Myricetin	2.38 × 10^4^	1.34 × 10^4^
Quinic acids	1.67 × 10^4^	1.72 × 10^4^
Hyperoside	2.19 × 10^5^	1.73 × 10^4^
Chlorophylls	4.89 × 10^4^	2.30 × 10^4^
Corosolic acid	2.83 × 10^4^	3.64 × 10^4^
Neochlorogenic acid	1.20 × 10^5^	3.66 × 10^4^
Ferulic acid	4.13 × 10^5^	5.29 × 10^4^
Phloretin	2133.20	6.76 × 10^4^
Oleic acid	1.11 × 10^6^	1.10 × 10^6^
Linoleic acid	1.77 × 10^6^	1.14 × 10^6^
Cinnamic acid	3.13 × 10^5^	3.99 × 10^6^

**Table 3 molecules-30-03978-t003:** Total phenolic content of different wild apple species.

Wild Apple Species	Total Phenolic Content, µg GAE/g FW	Study
*Malus baccata*	2033.33 †	[162,169,170]
*M. sieversii* f. *niedzwetzkyana*	2975.7 †	[90,115]
*Malus prunifolia*	8074–2.28 × 10^4^	[171]
*Malus sylvestris*	≈1600	[172]
*Malus sieversii*	477.84	[9]

Note: †—Mean value based on several studies.

## Data Availability

No new data were created in this study. Data sharing is not applicable.

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
