# Peer review of "The Biologically Active Compounds in Fruits of Cultivated Varieties and Wild Species of Apples"

_molecules, 2025, doi:10.3390/molecules30193978_

Round 1

Reviewer 1 Report

Comments and Suggestions for Authors
  1. The abstract needs major revision, especially the first sentence. Mentioning an exhaustive list of compounds is unnecessary in the abstract which should condense only the most important highlights of the paper. Moreover, there is no clear emphasis as to what the objectives of the review paper are. Although there is a mention between the comparison between wild and cultivated apples, it is not clear whether the review paper tackles this. The abstract should be a condensed summary of the major points discussed in the review.
  2. Section 2 describes the the active compounds commonly found in apples. As much as the discussion shows to be informative and comprehensive, covering its basic properties and their applications in biological/medical conditions. However, the it is too descriptive and it lacks some critical discussion; each subsection (2.1, 2.2, 2.3...) is too lengthy. It is suggested to present only the essential information that are relevant to the central focus of the review. 
  3. Alternatively, for all the subsections under Section 2, a short dicussion summarizing these compounds and a table that could clearly show the comparisons would be better. In the table, compound characteristics such as their properties, applications, etc. can be included. 
  4. Figure 7 has too much white space. Is it possible to have this figure rotated such that the bars are configured vertically? Or at least modify the figure to minimize the white space. Same goes for Figure 8.
  5. As for Section 3, the last paragraph (lines 882-894) discusses the comparison between the wild and cultivated apple species. The last sentence in particular states that wild species exhibit a more effective dose of the biologically active compounds compared to the cultivated species. Although the authors have stated that a comprehensive literature survey was done, there is still a lack of critical review. For example, line 887-891 talks about the amount of apple variety consumed in order to have a particular effective dose level, based on the given figures. However, it is still not clear as to how these results were obtained despite the mention of having done the literature survey.
  6. xfssd
  7. It is suggested that the overall paper be revised such that critical review should be included such as the comparison of results from various publications and what do these results imply in relation to the current review done by the authors.
  8. The conclusion is very much like an Introduction. It is too descriptive and does not state any real conclusion drawn from the discussions done in the previous sections. Moreover lines 942-949 enumerates the compounds that were mentioned in the previous sections - this is unnecessary to be included in the Conclusion. The conclusion section should only include the important information obtained from the discussions mentioned in the previous sections, it should address the objectives that are stated in the beginning of the review. With this, the paper also has a weak, if not lacking, clear objectives.
  9. Overall, this review paper was able to present substantial descriptive information with regards to biological compounds found in apples, but it needs a major revision and restructuring especially that the objectives are not clearly stated in the first place.

Reviewer 2 Report

Comments and Suggestions for Authors

This review offers a well-structured and insightful overview of the bioactive compounds present in apples, with a clear distinction between cultivated and wild varieties. The discussion on the implications for human health is relevant and well contextualized. I particularly appreciated the detailed examination of different compound classes, such as flavonoids, organic acids, pigments, and triterpenoids, which demonstrates a solid grasp of the subject.
The comparative analysis of compound levels between cultivated and wild apples is especially thought-provoking, reinforcing the idea that wild genotypes could be valuable sources of nutritional traits for breeding programs.
One point worth addressing is the image quality of the chemical structures, which could be improved to enhance readability and overall presentation.

Reviewer 3 Report

Comments and Suggestions for Authors

The manuscript by Shishparenok and co-authors reviews the diverse biologically active compounds found in apple fruits, such as flavonoids, triterpenoids, phenolic acids, and pigments. It highlights how traditional breeding for taste has inadvertently reduced the levels of these beneficial compounds. Moreover, the authors suggest that future breeding should prioritize increasing these health-promoting compounds in cultivated apple varieties. I believe the manuscript effectively integrates chemical profiles with biological functions, and offers valuable insight for future breeding strategies. The review is clearly written, scientifically sound, and suitable for publication in its current form.

Author Response

We appreciate the reviewer’s positive responses.

Reviewer 4 Report

Comments and Suggestions for Authors

This manuscript presents a comprehensive and well-structured comparative analysis of biologically active compounds in the fruits of cultivated apple varieties and wild species. The research is timely, scientifically sound, and relevant to the fields of plant science, food chemistry, and nutritional biology. The paper is well-written, methodologically rigorous, and contributes meaningfully to our understanding of the nutritional and phytochemical diversity within the Malus genus. This manuscript meets all the criteria for publication in its current form. It is a clearly presented review and I recommend it for publication.

Author Response

(The authors gave the same response as above.)

Reviewer 5 Report

Comments and Suggestions for Authors

Dear Authors,

The manuscript presents a comprehensive review of biologically active compounds in apple fruits, comparing cultivated varieties and wild species. The topic is relevant and timely, and the work offers a potentially significant contribution to the field by compiling extensive information on the health-related properties of polyphenols, organic acids, pigments, and triterpenoids. The breadth of coverage is impressive, and the review is generally well organized and supported by a substantial number of references to related and previous studies.

The scientific arguments are sound, with claims grounded in cited literature, and no misleading interpretations were observed. The overall structure is logical. The reference list is rich and appropriate.

In addition, all figures are of low resolution and should be improved for clarity and publication quality.

In summary, the work is scientifically sound and contributes meaningfully to the field, though its significance could be highlighted more strongly, and some refinements in organization, depth of explanation, and reference updating would improve the manuscript considerably.

Reviewer 6 Report

Comments and Suggestions for Authors

General Comments:

The manuscript, in its current form, lacks a clear structure and fails to provide a focused and critical analysis of the topic. The contribution of this review to the existing body of literature is not evident. Simply listing previously published studies with minimal commentary (typically 2–3 lines per paper) does not constitute a comprehensive or meaningful review article.

There is an absence of cohesive arguments and insightful discussion throughout the manuscript. The draft lacks depth in terms of conceptual explanation, mechanistic insights, and identification of research gaps. A review article should synthesize existing knowledge, highlight trends and controversies, and suggest directions for future research, none of which are sufficiently addressed here.

The manuscript appears disorganized and gives the impression of being randomly assembled. The introduction provides only general and vague background information, without clearly defining the scope, objectives, or novelty of the review. Similarly, the summary and conclusions are overly general, imprecise, and do not effectively consolidate the key findings or provide a strong takeaway message.

A more structured approach, organized into well-defined sub-sections and guided by a critical and thematic framework, is strongly recommended. Merely presenting literature data without analytical commentary does not fulfill the purpose of a scholarly review. Furthermore, a significant portion of the references cited are outdated and should be replaced with more recent and relevant studies.

Specific Comments:

  1. The novelty and rationale behind the review should be clearly articulated in the introduction.

  2. Authors should specify:

    • The databases used for literature search;

    • The keywords applied;

    • The inclusion/exclusion criteria for selecting the cited articles.

  3. The English language requires significant improvement for clarity and readability.

  4. All abbreviations should be defined upon first use.

  5. Typographical and grammatical errors are present throughout the manuscript and should be thoroughly corrected.

  6. Several references are outdated and should be replaced with more recent sources.

  7. Latin names in the references should follow standard formatting (italicized, with genus capitalized and species lowercase).

Recommendation:

Given the above concerns, the manuscript requires a major revision before it can be reconsidered for publication. A thorough reworking is necessary to improve the structure, language, analytical depth, and overall coherence of the review. At present, the manuscript does not meet the standards expected for publication in Molecules.

Reviewer 7 Report

Comments and Suggestions for Authors

The manuscript is a review of the biologically active compounds in fruits of cultivated varieties and wild species of apples.

The manuscript is interesting. Nevertheless, the manuscript should also present in more detail the main differences between the different cultivated varieties. Moreover, the active compounds should be presented concerning the respective apples rather than an extensive text with important information already presented in the literature.

The study should also present in more detail the advantages and drawbacks between the consumption of cultivated varieties and wild species of apples. It should also point out new directions.

Comments on the Quality of English Language

The English could be improved to more clearly express the research.

Round 2

Reviewer 1 Report

Comments and Suggestions for Authors

The authors made substantial revisions in correspondence to the initial comments. However, there are some points that needs minor revision:

  1. The numbers in the tables may be expressed better if in exponent form (e.g. 32058.89 change into 3.21 x 104 instead). It is neater to show the values this way.
  2. Is there any way for Table 1 to be shortened? Maybe change the column widths so that it won't appear too long in the manuscript.

Other than the above comments, no other suggestions are to be made. 

Reviewer 6 Report

Comments and Suggestions for Authors

The authors did not highlight all the changes they have made in the revised version of the manuscript, and without that, it is not possible to review the manuscript.

Reviewer 7 Report

Comments and Suggestions for Authors

The manuscript was improved.

Comments on the Quality of English Language

The English could be improved to more clearly express the research.

Round 3

Reviewer 6 Report

Comments and Suggestions for Authors

The authors answered all queries.